# Chemical reactivity theory to analyze possible toxicity of microplastics: Polyethylene and polyester as examples

Ana Martínez[1¤]*, Andrés Barbosa[2]

1 Departamento de Materiales de Baja Dimensionalidad, Instituto de Investigaciones en Materiales, Universidad Nacional Autónoma de México, CDMX, México, 2 Departamento de Ecología Evolutiva, Museo Nacional de Ciencias Naturales (MNCN-CSIC), Madrid, España

¤ Current address: Departamento de Ecología Evolutiva, Museo Nacional de Ciencias Naturales (MNCN-CSIC), Madrid, Spain
* martina@unam.mx

**Data Availability Statement:** All relevant data are within the manuscript.

**Funding:** The authors received no specific funding for this work.

## Abstract

Micro- and nanoplastics are widespread throughout the world. In particular, polyethylene (PE) and polyethylene terephthalate or polyester (PET) are two of the most common polymers, used as plastic bags and textiles. To analyze the toxicity of these two polymers, oligomers with different numbers of units were used as models. The use of oligomers as polymeric templates has been used previously with success. We started with the monomer and continued with different oligomers until the chain length was greater than two nm. According to the results of quantum chemistry, PET is a better oxidant than PE, since it is a better electron acceptor. Additionally, PET has negatively charged oxygen atoms and can promote stronger interactions than PE with other molecules. We found that PET forms stable complexes and can dissociate the guanine-cytosine nucleobase pair. This could affect DNA replication. These preliminary theoretical results may help elucidate the potential harm of micro- and nanoplastics.

## Introduction

Microplastics ($< 5$ mm in size) and nanoplastics ($< 100$ nm diameter) come from fragmentation of plastic particles through biological (metabolism), chemical (oxidation or hydrolysis) and physical degradation (UV interaction, mechanical processes) [1–6]. Micro- and nanoplastics are consumed by humans and animals. Previous research suggests that humans ingest an amount of plastics equivalent to one credit card per year [7, 8]. Micro- and nanoplastics are everywhere, but the question is whether they are really toxic.

There are many studies to determine possible danger of plastics in the environment and animal life [9–27]. Some indicate a relationship with oxidative stress. Others report that microplastics are ingested and evacuated without producing biochemical changes [22]. Recent investigations with Positron Emission Tomography to visualize biodistribution of radioplastics in mice reveal that most radioplastics remain in the gastrointestinal tract and, after 48 hours of consumption, they are eliminated through the feces [24]. While there is information for

**Competing interests:** The authors have declared that no competing interests exist.

humans and also for other species, all experiments reported so far used large concentrations of nanoplastics to test the effects [14]. The problem is that these concentrations cannot occur in the environment. Long-term exposure of environmental amounts is needed to understand the toxicity of nanoplastics in humans and other organisms. With all the information reported so far, the damage that micro- and nanoplastics can cause in humans or animals is still uncertain.

Polyethylene is the most common and cost-effective polymer, used as plastic bags and films [28–31]. Many kinds of polyethylene are known [28], with most having the chemical formula $(C_2H_4)_n$. The most common polyester is polyethylene terephthalate (PET). This has been extensively used in textiles. The demand of PET is increasing due to the extreme "use and waste" economy of clothing, promoted many times by fashion [29–31]. PET is also one of the most common polymers identified in samples of drinking water [11]. Nano- and microplastics of polyester and polyethylene could be dangerous.

Despite all the information we have about micro- and nanoplastics in different environments and about the effect that these particles can have on health, there is a lot of uncertainty about their harmful properties. There is also no theoretical research on potential toxicity of these polymers. There are some publications on molecular simulations to determine the effect of nanometric polystyrene particles [32] and the theory of chemical reactivity has been used to study the environmental risk [33–35] of different substances. There are also studies on oxidative stress and nanoplastics [36–39] but there are not theoretical studies on oxidative stress or the direct interactions of nanoplastics with DNA nitrogen bases. For this reason, the main idea of this investigation is to theoretically study different oligomers as models of polyethylene and polyester (polyethylene terephthalate, PET) using Density Functional Theory and different chemical reactivity indices (see Fig 1 for molecular formulas). Since polymers are difficult to optimize, oligomers with different numbers of units are used as models for polyethylene and polyester. Oligomers as models of polymers have been used previously with success [40, 41]. The results of this research can help to understand possible health effects of micro- and nanoplastics, and may determine which of these two plastics is potentially more dangerous.

## Computational details

Gaussian09 was used for all electronic calculations [42]. Geometry optimizations of initial geometries were obtained at M062x/6-311g+(2d, p) level of theory without symmetry constraints [43–45]. Harmonic analyses verified local minima (zero imaginary frequencies). Conceptual Density Functional Theory is a chemical reactivity theory founded on Density Functional Theory based concepts [46–53]. Within this theory there are response functions such as the electro-donating (ω-) and electro-accepting (ω+) powers, previously reported by

polyehtilene (PE)                    polyethylene terephthalate (PET)

**Fig 1. Schematic representation of PE and PET.**

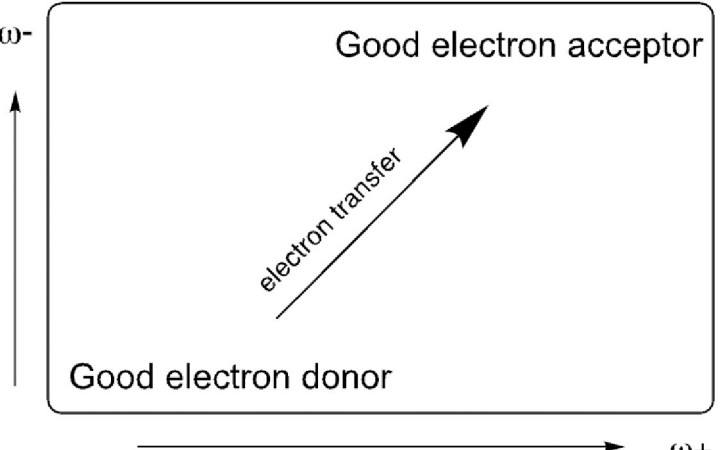

**Fig 2. Electron Donor-Acceptor Map (DAM).**

Gázquez *et al.* [48, 49] The propensity to donate electrons or ω- is defined as follows:

$$\omega- = (3I + A)^2/16(I - A) \tag{1}$$

whereas the propensity to accept electrons or ω+ is equal to

$$\omega+ = (I + 3A)^2/16(I - A) \tag{2}$$

I and A are vertical ionization energy and vertical electron affinity, respectively. They are obtained as follows:

$$A \rightarrow A^{+1} + 1e \quad I = E(A^{+1}) - E(A) \tag{3}$$

$$A^{-1} \rightarrow A + 1e \quad A = E(A) - E(A^{-1}) \tag{4}$$

Lower values of ω- indicate good electron donor molecules. Higher values of ω+ are for good electron acceptor molecules. ω- and ω+ refer to charge transfers, not necessarily from one electron. With these parameters it is possible to determine the Electron Donor-Acceptor Map (DAM, see Fig 2) [54]. Systems located down to the left are considered good electron donors whilst those situated up to the right are good electron acceptors. It can expect that electrons will be transferred from molecules considered good electron donors to those considered good electron acceptors. These chemical descriptors have been used successfully in many different chemical systems [55–59].

## Results and discussion

Different oligomers of polyester and polyethylene are used as models to investigate the electronic characteristics of polymers-like structures. This approach was previously used with success to study conducting polymers [40, 41]. Polymer biodegradation consists of several steps that break down large polymers to form the monomer, and then the monomer is mineralized to carbon dioxide and water [6, 60, 61]. Therefore, it is important to know the electronic properties as the size of the system decreases. For this reason, we started from the monomer and continued with different oligomers until the chain length was greater than 2 nm. In Figs 3 and

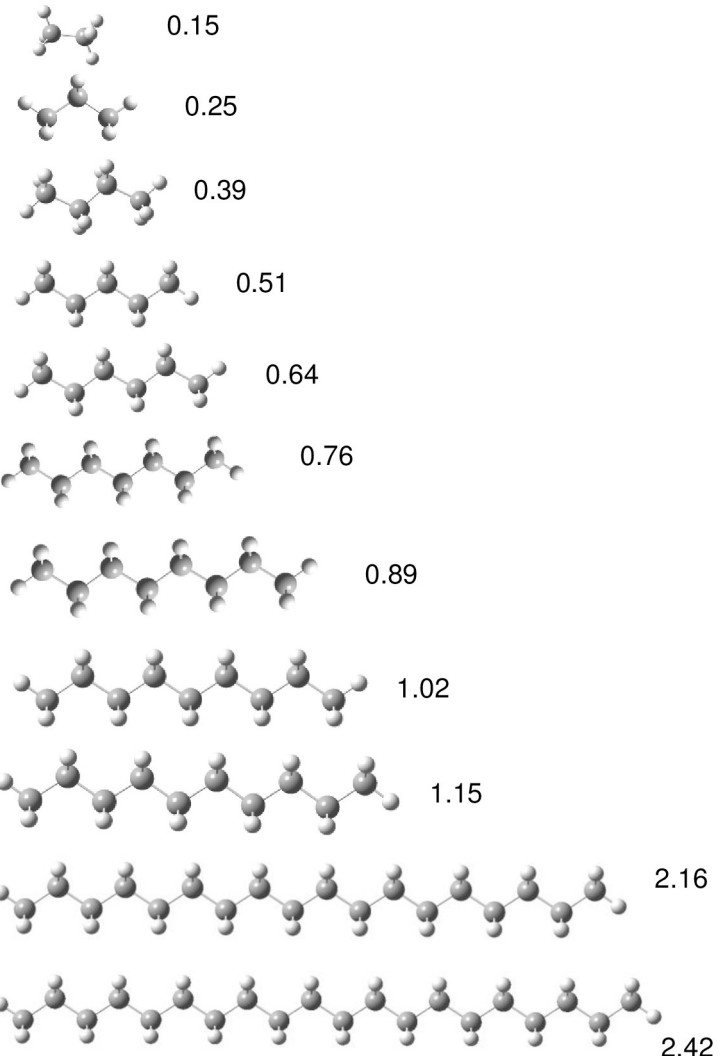

**Fig 3. Optimized structure of different models of polyethylene (PE).** Numbers represent chain length in nm.

4 the optimized structures of the models that investigated are shown. The correspondent length of the oligomer is also reported.

To investigate possible oxidative stress caused by micro- and nanoplastics, electro-donating and electro-accepting powers of all the systems under study were calculated. Good electron acceptors will take electrons from other systems. Good electron donors will donate electrons. The DAM for these oligomers is reported in Fig 5. Systems located down to the left are good electron donors. Therefore, they donate electrons producing the reduction of other molecules that gain these electrons. Systems located up the right are good electron acceptor. They accept electrons, oxidizing other species.

The results of Fig 5 show interesting patterns. For oligomers of polyethylene, the ability to donate and also the ability to accept electrons decreases as the size of the system increases. They become more reductant molecules and probably they are not capable of oxidizing other systems. For PET's oligomers the results are similar. The ability to accept or donate electrons of the three systems is different and there is also a correspondence with the size of the

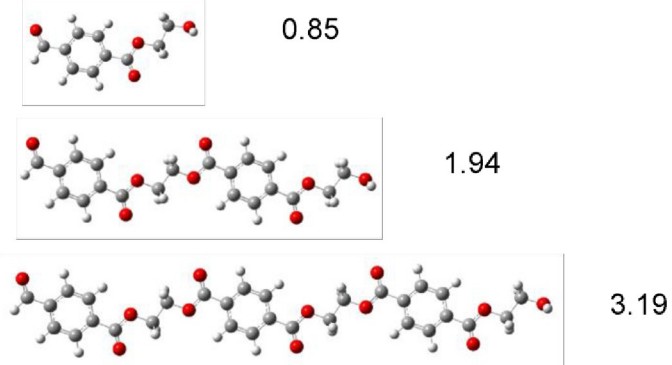

**Fig 4. Optimized structure of different models of polyester (PET).** Numbers represent chain length in nm.

oligomer. The bigger the system, the better electron acceptor it will be. They are located at the top right, so they are better electron acceptors and worse electron donors than the polyethylene´s oligomers. This result is as expected, since the electron affinity of oxygen is greater than that of carbon. The more oxygen atoms the molecule has, the better electron acceptor it will be. This means that PET´s oligomers are better oxidants than those of polyethylene and may produce oxidative stress oxidizing other molecules. Polyethylene´s oligomers are better electron donors and they may reduce other molecules. To investigate the importance of being a good oxidant or good reductant, it is necessary to compare with molecules of interest, as nucleobases.

To investigate the capability of oligomers to interact with nucleobases through electron transfer processes, we optimized geometries of adenine, thymine, cytosine and guanin, and we calculated the electron transfer properties. Adenine-thymine and cytosine-guanine pairs are

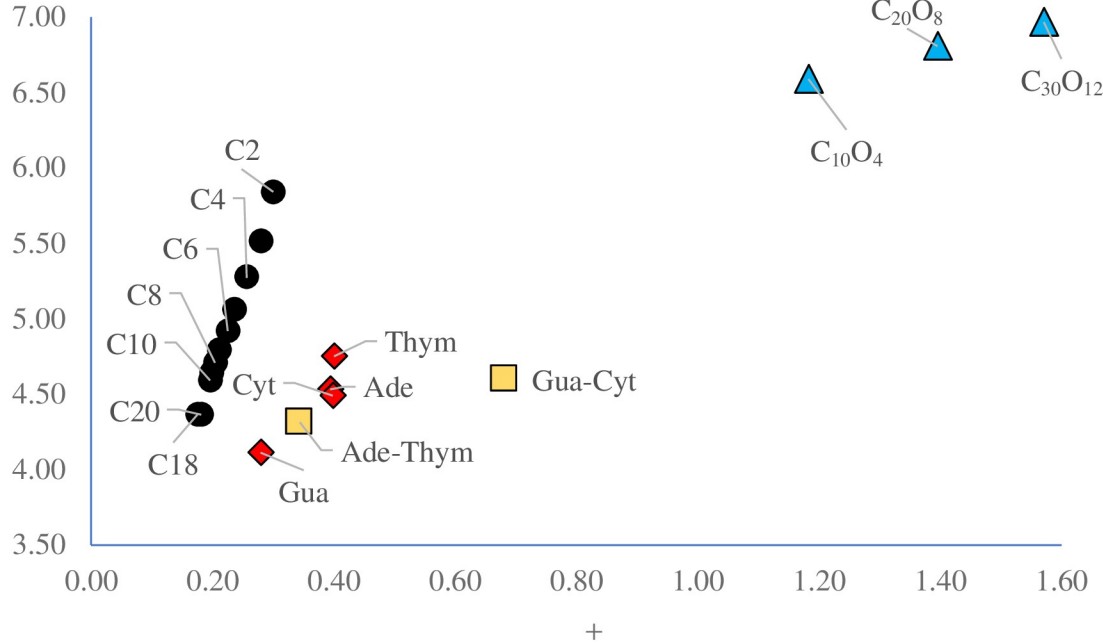

**Fig 5. DAM of system under study.** Values in eV.

also investigated. In order to compare the electron donor acceptor capacity of all systems. Results are reported in Fig 5. All nucleobases are located down to the left in the DAM. They are good electron donors and poor electron acceptors. Comparing with polyethylene´s oligomers, it can be seen that both are ubicated in the same region of the DAM. Electron donor capability is quite similar, being nucleobases slightly better electron acceptors. Therefore, no electron transfer is expected between polyethylene and nucleobases. With polyester the results are different. Nucleobases are located down to the left of the PET´s oligomers and electron transfer from nucleobases to these oligomers could be possible. Polyester can oxidize nucleobases and therefore, microplastics of PET can be harmful. More investigations are needed to corroborate this idea.

*In vivo* uptake and transport of nanoplastics depend on their own structure and properties, such as chemical composition [62]. Due to the differences in electronegativity of C and O, negative atomic charge in oxygen and carbon atoms can be anticipated. Mulliken Atomic Charges of $C_{18}H_{38}$ and $C_{20}H_{18}O_8$ as models of both nanoplastics corroborate this idea (see Fig 6). All oxygen atoms of PET oligomer are negatively charged whilst polyethylene oligomer presents negative carbon atoms. One possible risk of nanoplastics in the body is the interaction with important biomolecules as nucleobases for example. Both oligomers can form hydrogen bonds with nucleobases. PET can interact via oxygen atoms, while polyethylene can form hydrogen bonds with hydrogen atoms, since the carbons are sterically less accessible. The atomic charge of the hydrogens of polyethylene´s oligomer is smaller than the atomic charge of the oxygen atoms of PET´s oligomer. Stronger hydrogen bonds are expected with PET than with polyethylene.

To corroborate this idea and analyze possible harmful effects of nanoplastics, we investigated interactions of nucleobases with oligomers of polyethylene or polyester. The dissociation energies of X-guanine and X-guanine-cytosine (X is oligomers of polyethylene or PET) are analyzed to mimic possible interactions with DNA. The interactions of nanoplastics´ oligomers and guanine or guanine-cytosine base pair allow us to investigate possible effects of nanoplastics on these two nucleobases that are bonded in DNA. We used oligomers of

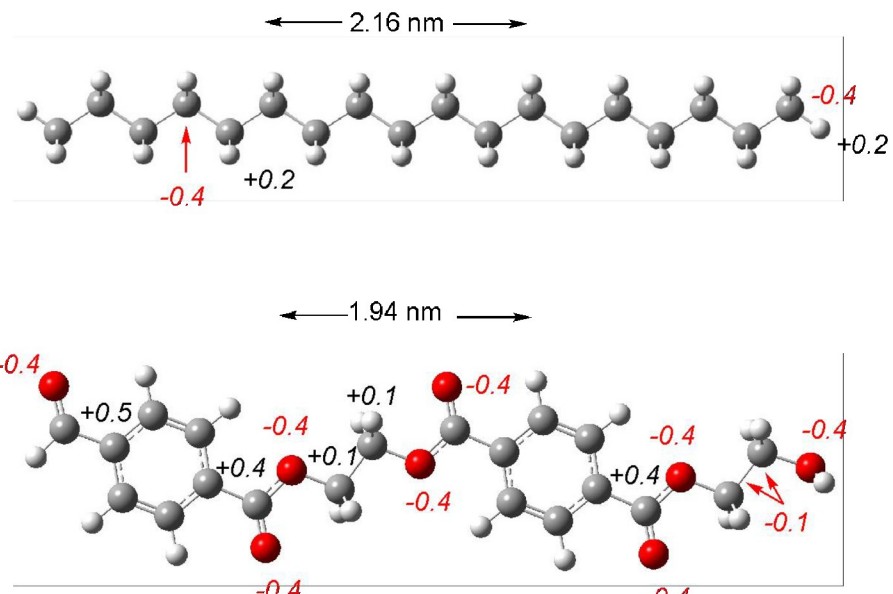

**Fig 6. Mulliken atomic charges of $C_{18}H_{38}$ and $C_{20}H_{18}O_8$.**

polyethylene and polyester, with ten and twenty carbon atoms respectively. Dissociation energies are calculated considering guanine or guanine-cytosine as products and following Eqs 5 and 6.

$$E_{dis} = \{[E(X) + E(guanine)] - E(X - guanine)\} \tag{5}$$

$$E_{dis} = \{[E(X) + E(guanine - cytosine)] - E(X - guanine - cytosine)\} \tag{6}$$

X is oligomers of polyethylene or PET. The results are reported in Fig 7. It is possible to see that different hydrogen bonds are formed. With polyethylene, the N and O of guanine form hydrogen bonds, with bond length of 2.2 to 2.8 Å. The hydrogen atoms of polyethylene form H-H bonds with cytosine and there is also one hydrogen bond with the oxygen atom of cytosine. The bond length of hydrogen bonds of cytosine is 2.0–2.6 Å. All PET- guanine and PET-guanine-cytosine hydrogen bonds are with an oxygen atom, either from PET, guanine or cytosine. The bond length of these hydrogen bonds is 2.0 to 2.3 Å. These results corroborate the idea that arises from the Mulliken atomic charges. Hydrogen atoms of polyethylene and oxygen atoms of PET form hydrogen bonds. As expected, oxygen atoms of guanine and cytosine also form hydrogen bonds with H atoms of polyethylene or PET. Due to the bond distance, it is expected that hydrogen bonds of nucleobases with PET are stronger than those of nucleobases with polyethylene. This conclusion is also obtained from the dissociation energies. Negative dissociation energies represent stable dissociated systems. Complexes with polyethylene´s oligomers are less stable than the dissociated system, *i.e.* stable complexes are not formed. With guanine, the binding energy is small (4.2 kcal/mol) and within the limits of the calculations, so the formation of a stable compound cannot be considered. For PET´s oligomers, the dissociation energy is positive and the complexes are more stable than the dissociated structures. PET's oligomer interacts with guanine, forming a 12.3 kcal more stable complex than the dissociated system. PET-guanine-cytosine complex is more stable than the dissociated system by 21.3 kcal/mol.

This could be related with the toxicity of micro- and nanoplastics. PET could be more dangerous than polyethylene since the interaction of correspondent oligomers with nucleobases is more stable for the first than for the second. Polyester forms stable complexes and can promote the dissociation of guanine-cytosine pairs. This possible interaction may be related to the toxicity of nanoplastics made from polyester. Polyethylene is expected to be less dangerous since the interaction with guanine and guanine-cytosine pair is not stable, so these nanoplastics will not interfere with DNA replication.

## Conclusions

Micro- and nanoplastics have long-term stability under environmental conditions, an important factor that increases the potential for living organisms to be exposed to these materials. So far, no clear toxic effects of micro and nano- have been observed. Based on the results reported here, it is possible to state that PET can be expected to be more harmful than PE for three reasons: PET is better electron acceptor and therefore a better oxidant than polyethylene; PET has negatively charged oxygen atoms and can promote stronger interactions than PE with other molecules; PET forms stable complexes and can dissociate the guanine-cytosine nucleobase pair. These first results contribute to understand potential dangerous of these two microplastics.

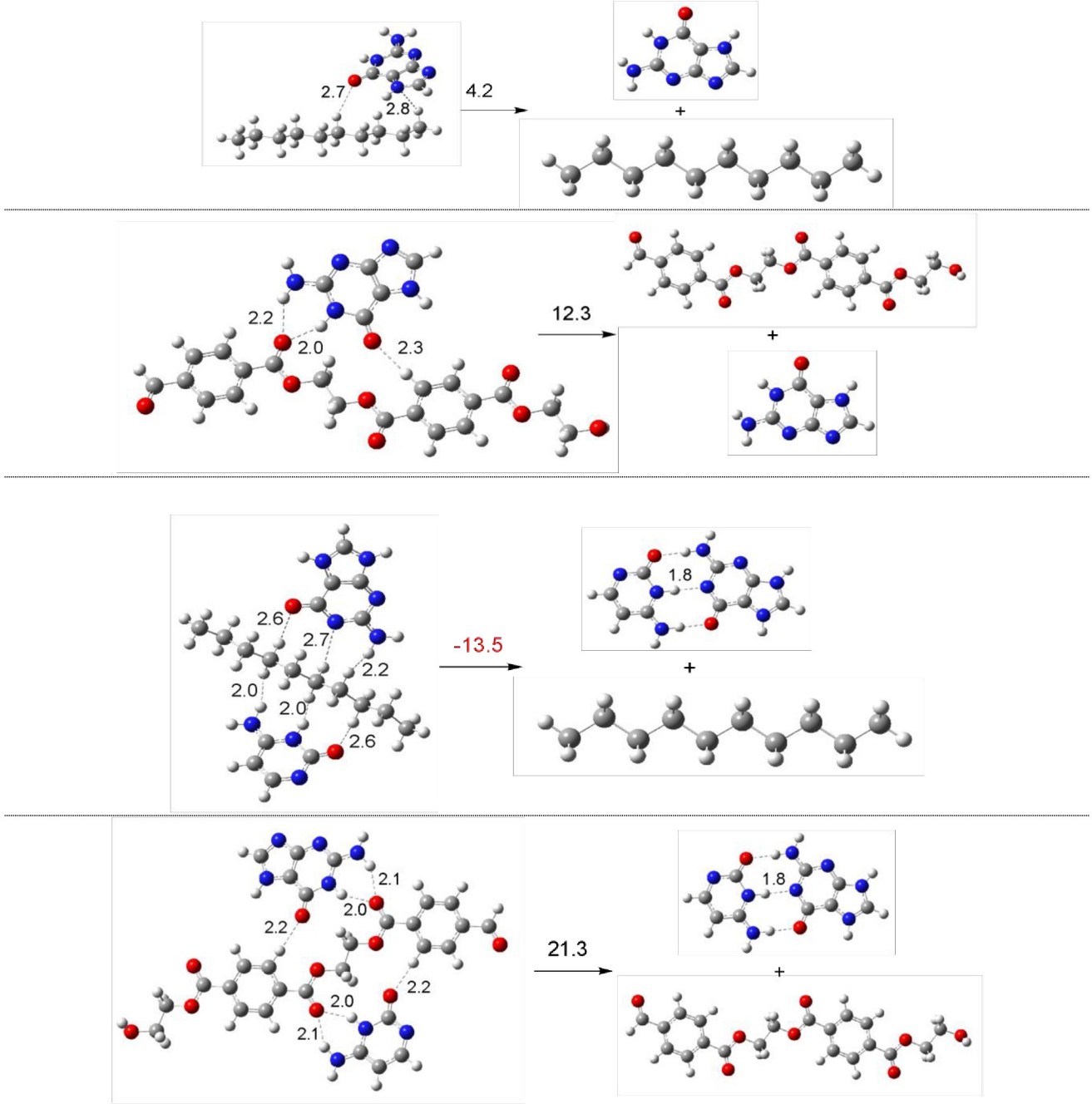

**Fig 7. Optimized structures and dissociation energies (in kcal/mol) of oligomers of polyethylene (C$_{10}$H$_{22}$) and PET (C$_{20}$H$_{18}$O$_8$) interacting with guanine and guanine-cytosine pair.** Dissociation Energies (E$_{dis}$, Kcal/mol) and corresponding chemical equation indicated with the optimized structures.

## Acknowledgments

AM acknowledges support from Universidad Nacional Autónoma de México and DGAPA through Programa de Apoyo para la Superación del Personal Académico de la UNAM (PASPA); and thanks to LANCAD-UNAM-DGTIC-141 for computer facilities.

## Author Contributions

**Conceptualization:** Ana Martínez, Andrés Barbosa.

**Formal analysis:** Ana Martínez.

**Investigation:** Ana Martínez, Andrés Barbosa.

**Methodology:** Andrés Barbosa.

**Writing – original draft:** Ana Martínez.

**Writing – review & editing:** Ana Martínez.

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
