## [Decision Letter · Decision Letter 0]

18 Sep 2023

PONE-D-23-12236Microplastics found in Antarctic penguins: chemical reactivity theory to analyze possible toxicityPLOS ONE

Dear Dr. Martínez,

Thank you for submitting your manuscript to PLOS ONE. After careful consideration, we feel that it has merit but does not fully meet PLOS ONE’s publication criteria as it currently stands. Therefore, we invite you to submit a revised version of the manuscript that addresses the points raised during the review process.

The manuscript needs extensive correction. I request the authors carefully read all the reviewers' comments and edit the manuscript  accordingly. .

We look forward to receiving your revised manuscript.

Kind regards,

Arumugam Sundaramanickam, PhD

Academic Editor

PLOS ONE

Journal Requirements:

"No. The funders had no role in study design, data collection and analysis, decision to publish, or preparation of the manuscript."

Reviewers' comments:

Reviewer's Responses to Questions

**Comments to the Author**

1. Is the manuscript technically sound, and do the data support the conclusions?

Reviewer #1: Yes

Reviewer #2: Partly

2. Has the statistical analysis been performed appropriately and rigorously? 

Reviewer #1: Yes

Reviewer #2: N/A

3. Have the authors made all data underlying the findings in their manuscript fully available?

Reviewer #1: Yes

Reviewer #2: Yes

4. Is the manuscript presented in an intelligible fashion and written in standard English?

Reviewer #1: Yes

Reviewer #2: Yes

5. Review Comments to the Author

Reviewer #1: Dear Editor,

Thank you for inviting me to review the current manuscript. I reviewed the manuscript and the manuscript need correction please check the following comments and points.

In the abstract, before stating the percentage of polymers, it is better to explain the sources of pollution in the region. Penguins were mentioned in the title, but no results were given regarding the number of microplastics and their size in the study area. How were polymers identified?

Fig 1- Which one is polyethylene and which one PET? Show in lowercase letters.

The discussion part of the results less compared with other studies.

The conclusion is very similar to the abstract.

Reviewer #2: The scope of this study is interesting and timing. Since we still do not know much about the real harm of micro- and nanoplastics continuously interacting with biological systems (from animals to humans), it is a research priority to predict their toxicity, as clearly stated in the introduction.

However, the authors focus on only two polymers found in penguin species (based on a single study) but they do not discuss why it is important in the context. The chemical reactivity theory can be applied to any context (I.e., any plastic polymer) and the results here can be associated to any organisms). What is the link to penguins rather than other species in which PE and PET have been found? If penguins are key in this study, the authors should first review all current literature on microplastic contamination in penguins and enlarge their study to depict a more comprehensive toxicity assessment.

Moreover, the authors discuss specific effects on DNA, which again can be referred to any biological system. What are the implications for penguin populations and Antarctic ecosystems? If the findings are too general, I suggest to simply remove penguins from the title.

The article readability can be improved by rechecking the grammar and trying to build more articulated sentences, which sometimes are very short. Some errors are reported below: (not having line numbers made the revision more difficult)

Additional comments:

Introduction

This study is based on citation n.27 (3 penguins species), but there are several studies on the subject, which should be at least mentioned on the introduction, then the authors should explain why they chose to focus only on findings from citation 27 (based on more samples? from more sampling sites?), as it is likely that penguins are not exposed to PE and PET only:

https://doi.org/10.1038/s41598-019-50621-2

https://doi.org/10.1038/s41598-023-39844-6

https://doi.org/10.1016/j.envint.2019.105303

No microplastic were also found in emperor penguins, this should be mentioned in the introduction as well: https://doi.org/10.1016/j.scitotenv.2022.158314

“There is also no theoretical research on potential toxicity of these polymers” There are model studies available on in vitro nanoplastic—cell interactions (e.g., https://doi.org/10.1021/jz402234c), although based on a different approach. I suggest listing some examples in which chemical reactivity theory has been applied to environmental risk assessment.

“The results of this research help…” change to: can help

“…and may determine which of the two plastics found in penguins is potentially more dangerous.” Please specify that you are referring to two types plastic polymers based one study.

…until the chain length is greater than 2 nm” change to: was

Rephrase to: In Figures 3 and 4 the optimized structures of the models investigated are shown. The

correspondent length of the oligomer is also reported.

“As was explained in the introduction, apparently micro- and nanoplastics may induce oxidative stress. To investigate possible oxidative stress, …” Rephrase to, deleting previous phrase: To investigate possible oxidative stress caused by micro- and nanoplastics,…

In page 6 there are some small errors to correct, please check the text again. Please avoiding the use of English possessive (‘s) with polymers.

“… we calculated adenine, thymine, cytosine and guanine.” In what sense? Add details

“The idea is to…” I suggest to change, continuing the previous sentence with as: in order to

“Polyester can oxidize nucleobases and therefore, microplastics of PET can be harmful. Although it was found that penguins have less polyester than polyethylene, nucleobases might transfer electrons to polyester and therefore it could affect more pinguin’s health.” Add details referred to this statement with actual examples of previous studies showing significant damages following exposure to PET microplastics to support your findings.

I suggest to add reference/s to page 9 to support the statements. The same applies to page 11: there are many in vitro/in vivo studies with model micro and nanoplastics, the authors should refer to those for comparison and to support their hypothesis.

Conclusions

“Micro- and nanoplastics can have long-term stability in various biological media (please add examples of biological media)

Change “obvious” to: clear

Change “say” to: state

“no toxic effects of micro and nano- have been observed”. I suggest to add: considering the diversity of biological systems.

“…for three reasons”. I suggest to list the reasons from the least to the most harmful, as in the abstract.

“Although it was found that penguins have…” change “have to: can ingest

As in this study only PE and PET are considered and no mention of the biological system I suggest the authors to refer to their results are first or preliminary.

6. PLOS authors have the option to publish the peer review history of their article (what does this mean?). If published, this will include your full peer review and any attached files.

Reviewer #1: No

Reviewer #2: No

---

## [Author Response · Author response to Decision Letter 0]

25 Sep 2023

PONE-D-23-12236

Old title: Microplastics found in Antarctic penguins: chemical reactivity theory to analyze possible toxicity.

New Title: Chemical reactivity theory to analyze possible toxicity of microplastics: polyethylene and polyester as examples.

Dear Arumugam Sundaramanickam, PhD

Academic Editor

PLOS ONE

Please find enclose the revised version of the manuscript entitled Chemical reactivity theory to analyze possible toxicity of microplastics: polyethylene and polyester as examples. In what follows we respond to each point raised by the academic editor and reviewers. All questions were answered to the best of our ability. 

We hope you find this version suitable for publication.

Kind regards,

Prof. Ana Martínez

UNAM

Reviewer #1: 

In the abstract, before stating the percentage of polymers, it is better to explain the sources of pollution in the region. Penguins were mentioned in the title, but no results were given regarding the number of microplastics and their size in the study area. How were polymers identified?

Author's response:

Following Reviewer #2's suggestion, we removed all information regarding penguins. Therefore, this information is not necessary.

Reviewer #1: 

Fig 1- Which one is polyethylene and which one PET? Show in lowercase letters.

Author's response:

Figure 1 was modified accordingly. 

Reviewer #1: 

The discussion part of the results less compared with other studies.

Author's response:

We modified the discussion accordingly

Reviewer #1: 

The conclusion is very similar to the abstract.

Author's response:

We modified the abstract as follows:

Micro- and nanoplastics are widespread throughout the world. In particular, polyethylene (PE) and polyethylene terephthalate or polyester (PET) are two of the most common polymers, used as plastic bags and textiles. To analyze the toxicity of these two polymers, oligomers with different numbers of units were used as models. The use of oligomers as polymeric templates has been used previously with success. We started with the monomer and continued with different oligomers until the chain length was greater than two nm. According to the results of quantum chemistry, PET is a better oxidant than PE, since it is a better electron acceptor. Additionally, PET has negatively charged oxygen atoms and can promote stronger interactions than PE with other molecules. We found that PET forms stable complexes and can dissociate the guanine-cytosine nucleobase pair. This could affect DNA replication. These preliminary theoretical results may help elucidate the potential harm of micro- and nanoplastics.

Reviewer #2: 

However, the authors focus on only two polymers found in penguin species (based on a single study) but they do not discuss why it is important in the context. The chemical reactivity theory can be applied to any context (I.e., any plastic polymer) and the results here can be associated to any organisms). What is the link to penguins rather than other species in which PE and PET have been found? If penguins are key in this study, the authors should first review all current literature on microplastic contamination in penguins and enlarge their study to depict a more comprehensive toxicity assessment. Moreover, the authors discuss specific effects on DNA, which again can be referred to any biological system. What are the implications for penguin populations and Antarctic ecosystems? If the findings are too general, I suggest to simply remove penguins from the title.

Author's response:

We agree with this statement. Penguins are not key to this research. We included them because one of us found microplastics in penguins, but this is no reason to preserve the idea of penguins. In this new version we eliminate all information related to penguins.

Reviewer #2: 

The article readability can be improved by rechecking the grammar and trying to build more articulated sentences, which sometimes are very short. Some errors are reported below: (not having line numbers made the revision more difficult)

Author's response:

We corrected the errors and review the grammar from the beginning. 

Reviewer #2: 

This study is based on citation n.27 (3 penguins species), but there are several studies on the subject, which should be at least mentioned on the introduction, then the authors should explain why they chose to focus only on findings from citation 27 (based on more samples? from more sampling sites?), as it is likely that penguins are not exposed to PE and PET only: 

https://doi.org/10.1038/s41598-019-50621-2

https://doi.org/10.1038/s41598-023-39844-6

https://doi.org/10.1016/j.envint.2019.105303

No microplastic were also found in emperor penguins, this should be mentioned in the introduction as well: https://doi.org/10.1016/j.scitotenv.2022.158314

Author's response:

Following the suggestion of this reviewer, we delete all information about pinguins. 

Reviewer #2: 

“There is also no theoretical research on potential toxicity of these polymers” There are model studies available on in vitro nanoplastic—cell interactions (e.g., https://doi.org/10.1021/jz402234c), although based on a different approach. I suggest listing some examples in which chemical reactivity theory has been applied to environmental risk assessment.

Author's response:

Following this suggestion, we added more references and the following paragraph:

There are some publications on molecular simulations to determine the effect of nanometric polystyrene particles [32] and the theory of chemical reactivity has been used to study the environmental risk [33-35] of different substances, but there are no studies on oxidative stress or the direct interactions of nanoplastics with DNA nitrogen bases. For this reason, …

New references:

32. Rossi G, Barnoud J, Monticelli L. Polystyrene Nanoparticles Perturb Lipid Membranes 

J. Phys. Chem. Lett. 2014; 5, 241−246. dx.doi.org/10.1021/jz402234c

33. Wang Q, Zhang Y, Rogers WJ, Mannan MS. Molecular simulation studies on chemical reactivity of methylcyclopentadiene. Journal of Hazardous Materials 2009; 165, 141–147 doi: 10.1016/j.jhazmat.2008.09.087 

34. Rawat P, Singh RN. Experimental and theoretical study of 4-formyl pyrrole derived aroylhydrazones. Journal of Molecular Structure 2015; 1084, 326–339 http://dx.doi.org/10.1016/j.molstruc.2014.12.045

35. Villaverde JJ López-Goti C, Alcamí M, Lamsabhi AM, Alonso-Pradosa JL, Sandín-España P. Quantum chemistry in environmental pesticide risk assessment. Pest. Manag. Sci. 2017; 73: 2199–2202 doi: 10.1002/ps.4641 

Reviewer #2: 

“The results of this research help…” change to: can help

“…and may determine which of the two plastics found in penguins is potentially more dangerous.” Please specify that you are referring to two types plastic polymers based one study.

…until the chain length is greater than 2 nm” change to: was

Rephrase to: In Figures 3 and 4 the optimized structures of the models investigated are shown. The correspondent length of the oligomer is also reported.

“As was explained in the introduction, apparently micro- and nanoplastics may induce oxidative stress. To investigate possible oxidative stress, …” Rephrase to, deleting previous phrase: To investigate possible oxidative stress caused by micro- and nanoplastics,…

In page 6 there are some small errors to correct, please check the text again. Please avoiding the use of English possessive (‘s) with polymers.

“The idea is to…” I suggest to change, continuing the previous sentence with as: in order to. “Polyester can oxidize nucleobases and therefore, microplastics of PET can be harmful. 

Change “obvious” to: clear

Change “say” to: state

“no toxic effects of micro and nano- have been observed”. I suggest to add: considering the diversity of biological systems.

“…for three reasons”. I suggest to list the reasons from the least to the most harmful, as in the abstract.

“Although it was found that penguins have…” change “have to: can ingest 

Author's response:

We made all these correction in this new version. We delete all information concerning penguins.

Reviewer #2: 

“… we calculated adenine, thymine, cytosine and guanine.” In what sense? Add details

Author's response:

To answer this question, we modified the paragraph as follows:

we optimized geometries of adenine, thymine, cytosine and guanin, and we calculated the electron transfer properties.

Reviewer #2: 

Although it was found that penguins have less polyester than polyethylene, nucleobases might transfer electrons to polyester and therefore it could affect more pinguin’s health.” Add details referred to this statement with actual examples of previous studies showing significant damages following exposure to PET microplastics to support your findings.

Author's response:

We delete all information about pinguins so this correction is not necessary. 

Reviewer #2: 

I suggest to add reference/s to page 9 to support the statements. The same applies to page 11: there are many in vitro/in vivo studies with model micro and nanoplastics, the authors should refer to those for comparison and to support their hypothesis.

Author's response:

We performed a review of the literature and found no evidence on the comparison of toxicity of PP and PET. I'm afraid I don't understand what evidence the reviewer is suggesting, but I'm interested in following this suggestion as soon as I get the idea.

Reviewer #2: 

“Micro- and nanoplastics can have long-term stability in various biological media (please add examples of biological media)

Author's response:

This is a mistake. We were thinking on the environment. The new sentence is as follows:

“Micro- and nanoplastics can have long-term stability under environmental conditions.

Reviewer #2: 

As in this study only PE and PET are considered and no mention of the biological system, I suggest the authors to refer to their results are first or preliminary.

Author's response:

We added this sentence at the end of the abstract and conclusions.

---

## [Decision Letter · Decision Letter 1]

15 Nov 2023

PONE-D-23-12236R1Chemical reactivity theory to analyze possible toxicity of microplastics: polyethylene and polyester as examplesPLOS ONE

Dear Dr. Martinez Vazquez,

Thank you for submitting your manuscript to PLOS ONE. After careful consideration, we feel that it has merit but does not fully meet PLOS ONE’s publication criteria as it currently stands. Therefore, we invite you to submit a revised version of the manuscript that addresses the points raised during the review process.

We look forward to receiving your revised manuscript.

Kind regards,

Arumugam Sundaramanickam, PhD

Academic Editor

PLOS ONE

Journal Requirements:

Reviewers' comments:

Reviewer's Responses to Questions

**Comments to the Author**

1. If the authors have adequately addressed your comments raised in a previous round of review and you feel that this manuscript is now acceptable for publication, you may indicate that here to bypass the “Comments to the Author” section, enter your conflict of interest statement in the “Confidential to Editor” section, and submit your "Accept" recommendation.

Reviewer #1: All comments have been addressed

Reviewer #2: All comments have been addressed

2. Is the manuscript technically sound, and do the data support the conclusions?

Reviewer #1: Yes

Reviewer #2: Yes

3. Has the statistical analysis been performed appropriately and rigorously? 

Reviewer #1: Yes

Reviewer #2: N/A

4. Have the authors made all data underlying the findings in their manuscript fully available?

Reviewer #1: Yes

Reviewer #2: Yes

5. Is the manuscript presented in an intelligible fashion and written in standard English?

Reviewer #1: Yes

Reviewer #2: Yes

6. Review Comments to the Author

Reviewer #1: Authors have responded effectively to the initial querries raised. Since the necessary corrections have been made, it is appropriate to publish it as it is.

Reviewer #2: "...there are no studies on oxidative stress or the direct interactions of nanoplastics with DNA nitrogen bases"

this should be changed as there are several studies on ox stress and nanoplastics, for example: https://doi.org/10.1016/j.envint.2023.107899

https://doi.org/10.3389/fpls.2022.1027608
https://doi.org/10.3390/antiox11020193, and as reviewed in https://doi.org/10.1016/j.scitotenv.2023.163144

Regarding the references to add to support these statements:

"In vivo uptake and transport of nanoplastics depend on their own structure and properties,

such as chemical composition" the authors can easily refer to reviews summarising these concepts as https://

doi.org/10.3390/nano11081903

All other comments have been addressed

7. PLOS authors have the option to publish the peer review history of their article (what does this mean?). If published, this will include your full peer review and any attached files.

Reviewer #1: **Yes: **Mohammad Gholizadeh

Reviewer #2: No

---

## [Author Response · Author response to Decision Letter 1]

15 Nov 2023

Chemical reactivity theory to analyze possible toxicity of microplastics: polyethylene and polyester as examples.

Dear Arumugam Sundaramanickam, PhD

Academic Editor

PLOS ONE

Please find enclose the revised version of the manuscript entitled Chemical reactivity theory to analyze possible toxicity of microplastics: polyethylene and polyester as examples. In what follows we respond to each point raised by the academic editor and reviewers. All questions were answered to the best of our ability. 

We hope you find this version suitable for publication.

Kind regards,

Prof. Ana Martínez

UNAM

Reviewer #1: 

Authors have responded effectively to the initial querries raised. Since the necessary corrections have been made, it is appropriate to publish it as it is.

Reviewer #2: 

"...there are no studies on oxidative stress or the direct interactions of nanoplastics with DNA nitrogen bases"

this should be changed as there are several studies on ox stress and nanoplastics, for example: https://doi.org/10.1016/j.envint.2023.107899;
https://doi.org/10.3389/fpls.2022.1027608

https://doi.org/10.3390/antiox11020193, and as reviewed in https://doi.org/10.1016/j.scitotenv.2023.163144

Author's response:

They are very interesting references, which are now included in this new version. The following paragraph is also added.

There are also studies on oxidative stress and nanoplastics [36-40] but there are not theoretical studies on oxidative stress or the direct interactions of nanoplastics with DNA nitrogen bases.

36. Ekner-Grzyb A, Duka A, Grzyb T, Lopes I, Chmielowska-Ba ˛k J. Plants oxidative response to nanoplastic. Front. Plant Sci. 2022, 13:1027608. doi: 10.3389/fpls.2022.1027608 

37. Ferrante MC, Monnolo A, Del Piano F, Mattace Raso G, Meli R. The pressing issue of micro- and nanoplastic contamination: profiling the reproductive alterations mediated by oxidative stress. Antioxidants 2022, 11: 193 https://doi.org/10.3390/ antiox11020193 

38. Ding R, Ma Y, Li T, Sun M, Sun Z, Duan J. The detrimental effects of micro-and nano-plastics on digestive system: An overview of oxidative stress-related adverse outcome Science Tot. Env. 2023, 878: 163144. https://doi.org/10.1016/j.scitotenv.2023.163144

39. Zhou Y, He G, Jiang H, Pan K, Liu W. Nanoplastics induces oxidative stress and triggers lysosome-associated immune-defensive cell death in the earthworm Eisenia fetida. Env. Int. 2023, 174: 107899. https://doi.org/10.1016/j.envint.2023.107899.

Regarding the references to add to support these statements:

"In vivo uptake and transport of nanoplastics depend on their own structure and properties,

such as chemical composition" the authors can easily refer to reviews summarising these concepts as https://

doi.org/10.3390/nano11081903

The reference is already added as follows:

62. Corsi I, Bellingeri A, Eliso MC, Grassi G, Liberatori G, Murano C, Sturba L, Vannuccini ML, Bergami E. Eco-interactions of engineered nanomaterials in themarine Environment: towards an eco-design framework. Nanomaterials 2021, 11: 1903. doi.org/10.3390/nano11081903

The reference number was also updated and indicated in red.

---

## [Decision Letter · Decision Letter 2]

2 Feb 2024

Chemical reactivity theory to analyze possible toxicity of microplastics: polyethylene and polyester as examples

PONE-D-23-12236R2

Dear Dr. Martinez Vazquez,

We’re pleased to inform you that your manuscript has been judged scientifically suitable for publication and will be formally accepted for publication once it meets all outstanding technical requirements.

Kind regards,

Arumugam Sundaramanickam, PhD

Academic Editor

PLOS ONE
---

## [Editor Report · Acceptance letter]

13 Feb 2024

PONE-D-23-12236R2 

PLOS ONE

Dear Dr. Martínez, 

I'm pleased to inform you that your manuscript has been deemed suitable for publication in PLOS ONE. Congratulations! Your manuscript is now being handed over to our production team.

Kind regards, 

on behalf of

Professor Arumugam Sundaramanickam 

Academic Editor

PLOS ONE